# Shape from Blur: Recovering Textured 3D Shape and Motion of Fast Moving Objects

**Denys Rozumnyi**
Department of Computer Science
ETH Zurich, Switzerland
denys.rozumnyi@inf.ethz.ch

**Martin R. Oswald**[1,2]
[2]University of Amsterdam
Netherlands
martin.oswald@inf.ethz.ch

**Vittorio Ferrari**
Google Research
Zurich, Switzerland
vittoferrari@google.com

**Marc Pollefeys**
[1]Department of Computer Science
ETH Zurich, Switzerland
marc.pollefeys@inf.ethz.ch

## Abstract

We address the novel task of jointly reconstructing the 3D shape, texture, and motion of an object from a single motion-blurred image. While previous approaches address the deblurring problem only in the 2D image domain, our proposed rigorous modeling of all object properties in the 3D domain enables the correct description of arbitrary object motion. This leads to significantly better image decomposition and sharper deblurring results. We model the observed appearance of a motion-blurred object as a combination of the background and a 3D object with constant translation and rotation. Our method minimizes a loss on reconstructing the input image via differentiable rendering with suitable regularizers. This enables estimating the textured 3D mesh of the blurred object with high fidelity. Our method substantially outperforms competing approaches on several benchmarks for fast moving objects deblurring. Qualitative results show that the reconstructed 3D mesh generates high-quality temporal super-resolution and novel views of the deblurred object.

## 1 Introduction

Motion blur is a common cause of degraded image quality. It can originate from camera motion, fast object motion, long exposure times due to low light settings, or a combination of these effects. In this paper, we focus on deblurring of fast moving objects (FMOs), which move over a distance larger than their size for the duration of the camera exposure of a single image. Since FMOs appear as a mixture of motion-blurred object texture and the background, our first goal is to decompose the combined appearance of the background and the one of the FMO in form of a matting mask [1]. Our main goal is to recover the shape, motion, and sharp texture of the FMO in order to explain its appearance in the input image in the best possible way (Fig. 1). The accurate reconstruction and motion estimation of FMOs enables to generate videos with temporal super-resolution, as well as FMO motion analysis and prediction. This is useful for a variety of applications like sports analysis (e.g. soccer, tennis, baseball), meteorites detection in astrophysics, detecting fast moving obstacles in front of a vehicle for driver alert or autonomous driving, and general video quality enhancement or compression. Our work could eventually make FMO tracking accessible to everyone using regular cameras rather than expensive high-speed cameras, or push the capabilities of current high-speed cameras to a new level, e.g. by capturing gun shots.

The majority of deblurring methods aim for the generic task of removing any kind of blur in images [2, 3, 4] or videos [5, 6, 7, 8, 9]. However, they perform poorly on deblurring of FMOs as

35th Conference on Neural Information Processing Systems (NeurIPS 2021).

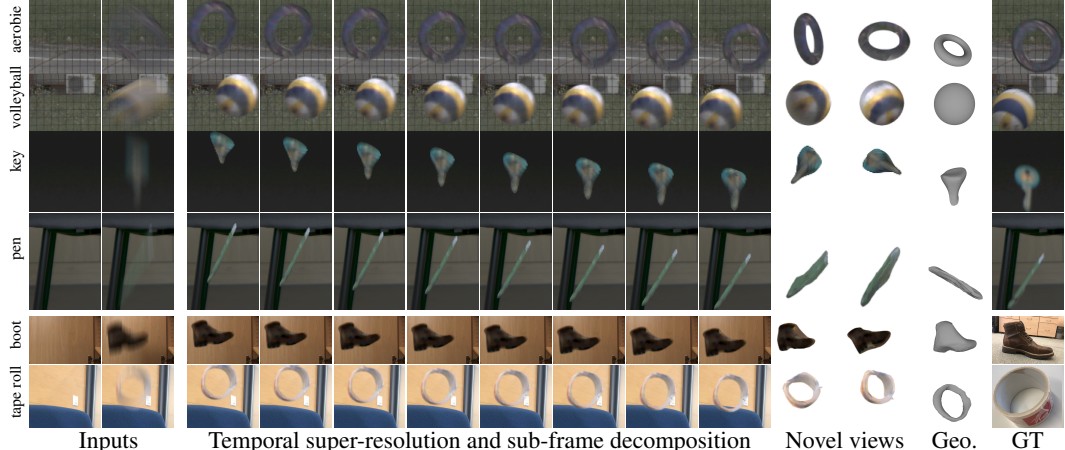

| aerobie | volleyball | key | pen | boot | tape roll |

Inputs     Temporal super-resolution and sub-frame decomposition     Novel views    Geo.    GT

Figure 1: **Temporal super-resolution and deblurring with 3D reconstruction of fast moving objects, given a single image.** Solely from the inputs on the left (clean background $B$, blurry image $I$), the proposed Shape-from-Blur (SfB) method reconstructs a high-quality textured 3D shape and its motion, by minimizing the input image reconstruction loss with suitable regularizers. Our method enables to generate temporal super-resolution from a single image, as well as synthesizing novel views of the reconstructed shape. The ground truth (GT) on the right shows the corresponding high-speed camera footage (rows 1-4) or a picture of the object without motion blur (rows 5-6).

shown in comparisons to methods specialized for them [10, 11]. To the best of our knowledge, all generic and all FMO-specialized methods try to tackle the deblurring task only in the 2D image domain. This limits their ability to describe object rotations around any axis that does not correspond to the camera viewing direction, or any object translation along the viewing direction.

Our key idea is to model the object shape, motion, and texture in the 3D domain to better explain the input image as the projection and temporal aggregation of a continuously moving object. We recover motion as the initial 6-DoF pose of the object and the 3D translation and 3D rotation that it undergoes during the exposure time of the input image. Moreover, the proposed method is the first to estimate the 3D shape of motion-blurred objects. Although we are solving a more general and more challenging problem than previous methods, we achieve substantially better deblurring results due to the rigorous joint modeling of object shape, motion, and texture in 3D.

In summary, we make the following **contributions: (1)** We propose a novel fast moving object deblurring method that for the first time jointly estimates from a single input image the 3D shape, texture, and motion of an object (initial 6-DoF pose, 3D translation and 3D rotation). **(2)** We describe an effective pipeline that uses pure test-time optimization for the recovery of the object 3D shape, motion, and sharp appearance. To constrain the optimization, we propose several suitable regularizers.

## 2  Related work

**Generic deblurring.**  Single-image deblurring is studied in [12, 2], and video deblurring in [9, 5, 6, 7, 8]. Some methods were specifically proposed for motion deblurring [13, 14, 15], but they generate only a single output image or a motion flow. A more difficult task of estimating the temporal super-resolution from a single image has been addressed in [3, 4, 16], but only for small motions. Below we review methods specialized to deblurring fast moving objects (FMOs).

**Constant 2D object appearance.**  The image formation with motion blur of a FMO [10, 17] was introduced as

$$I = H * F + (1 - H * M) \cdot B \ , \tag{1}$$

where the input image $I$ is formed as a composition of two terms. The first term represents the blurred object as a convolution of the sharp object appearance $F$ and the blur kernel $H$. The second term encodes the influence of the background $B$, which depends on the object matting mask $M$. This model assumes that the object appearance is constant throughout the whole exposure time. Tracking by Deblatting (TbD) [18, 19], from deblurring and matting, was proposed to estimate the unknown

sharp object model $(F, M)$ and its motion represented by $H$. To achieve this, Kotera et al. [18] applied ADMM variable splitting with appearance total variation and blur kernel sparsity priors. Follow-up methods improve the precision of the 2D trajectory estimation [20, 21]. The special case of planar object with planar rotation is studied in [22].

**Piece-wise constant 2D object appearance.** The constant appearance model (1) was extended to piece-wise constant [23] to deal with small amounts of appearance changes over time. The appearance might change over time due to object translation in 3D, object rotation in 3D, view-specific changes due to illumination, specularities, shadows, and even object deformation, e.g. when the object bounces off the ground. The piece-wise constant model is defined as

$$I = \sum_\tau H_\tau * F_\tau + \left(1 - \sum_\tau H_\tau * M_\tau\right) \cdot B \ , \tag{2}$$

where the exposure time is split into several uniform time intervals $\tau$ (sub-frames). Each sub-frame has its own appearance model $(F_\tau, M_\tau)$ and blur kernel $H_\tau$, which are constant within the sub-frame. TbD-3D [23] proposed to estimate these piece-wise appearances by energy minimization with several priors, such as rotational symmetry, appearance total variation, and penalizing appearance from changing rapidly across neighboring sub-frames. The blur kernel is still estimated by deblatting [18] assuming the simple constant appearance model (1), and the sub-frame appearances are estimated in a second pass by splitting the trajectory into the desired number of pieces. Thus, TbD-3D performs well only with very small appearance changes and mostly spherical objects due to the rotational symmetry prior. For perfectly spherical objects, TbD-3D can also estimate 3D rotation by minimizing the sphere reprojection error. Then, the object-camera distance is approximated by the radius of estimated 2D masks. However, this method cannot generalize to other shapes.

**2D projections of any object.** DeFMO [11] was the first learning-based method for FMO deblurring. They generalized the image formation model further to

$$I = \int_0^1 F_\tau \cdot M_\tau \, d\tau + \left(1 - \int_0^1 M_\tau \, d\tau\right) \cdot B \ , \tag{3}$$

where the sharp object appearance $F_\tau$ is defined at any real-valued time point $\tau$ as an output of their encoder-decoder network architecture. The integration bounds correspond to the normalized image exposure time with start and end times 0 and 1, respectively. In (3), the 2D object motion is directly encoded in the masks $M_\tau$. The network is trained end-to-end on a synthetic dataset with various moving 3D objects. DeFMO generalizes well to real-world data, but the outputs are still just 2D projections of the moving object. There is no guarantee that they correspond to the object's actual 3D motion and shape, and there is no straightforward way to estimate the 3D shape from 2D DeFMO outputs. The 3D motion is not estimated either.

**3D shape from sharp image.** Estimating 3D shape from a single sharp image is a hot topic in computer vision. Recent methods are based on differentiable or neural rendering [24, 25, 26, 27] and Neural Radiance Fields (NeRF) [28, 29, 30]. Many of those methods heavily depend on shape priors and/or large datasets of training images (of an object class [26, 27, 30, 24, 25], sometimes even multiple views of the same object [28, 29]). Over the last 30 years, a variety of methods have been proposed for monocular 3D shape estimation from other clues such as Shape from Shading [31], Shape from Texture [32], Shape from Focus [33], Shape from Defocus [34], Shape from Shadow [35], Shape from Specularities [36], and others. We augment this family of "Shape from X" methods with a new Shape from Blur (SfB). To some degree, SfB can be related to Shape from Motion (SfM) [37] and Shape from Silhouettes [38], but in our case, the 2D image sequence is collapsed and averaged into a single image by motion blur. In contrast to all previous FMO deblurring methods, we explicitly estimate textured 3D shape and 3D motion of such objects. In contrast to other 3D shape estimation methods, we estimate the 3D shape of an object with significant motion blur from a single image (plus the background).

## 3 Method

The input to our method is a single RGB image $I$ of a significantly blurred fast moving object. Additionally, an approximate image of the background $B$ without FMOs is given. This can be estimated as a median of consecutive video frames or as a static image without the object. Hence, the

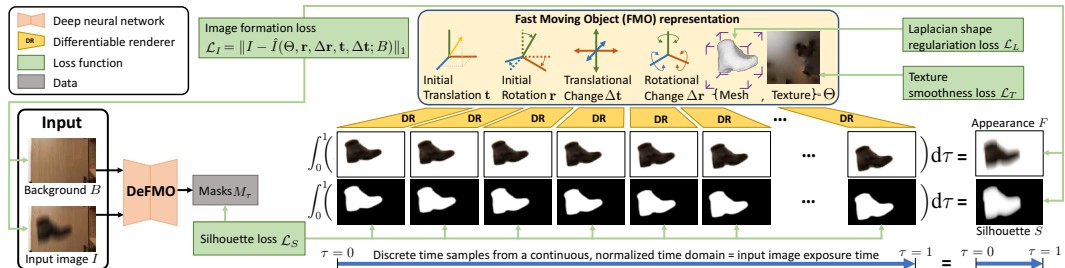

Figure 2: **Overview of our method.** Given an input image and a clean background, together with the matting masks produced by DeFMO [11], our method estimates the 3D shape, texture, and motion of the object. This is achieved by jointly minimizing four suitable loss functions.

assumption of known background is not very restrictive and also common among all FMO-related methods [10, 18, 23, 11]. Our method outputs a textured 3D mesh $\Theta$ of the object and its sub-frame motion, represented as its initial 6-DoF pose $\{\mathbf{r}, \mathbf{t}\}$, and the 3D translation $\Delta\mathbf{t}$ and 3D rotation $\Delta\mathbf{r}$ it undergoes during the image exposure time. The key idea of our method is to reconstruct the textured 3D shape and motion parameters that best explain the input image. Concretely, we render the reconstructed object with its motion blur and define a loss comparing the resulting image with the input (Fig. 2). We then find the texture, shape, and motion parameters that minimize this loss.

**Shape representation.** We represent the shape and appearance of an object by a textured triangular mesh $\Theta$, which consists of a set of vertices, a set of faces, and a texture map with a given texture mapping. The mesh topology (the set of faces and their connectivity) is fixed. Our method estimates offsets from the initial vertex positions of a given prototype shape to deform it into the 3D shape of the object in the input image. We automatically select a prototype shape out of a given set based on the quality of the rendered image (see Sec. 3.2).

**Image formation model.** We define two rendering functions $\mathcal{R}_F$ and $\mathcal{R}_S$ that output the projected object appearance $F = \mathcal{R}_F(\Theta)$ and the object silhouette $S = \mathcal{R}_S(\Theta)$ as 2D images from a given mesh $\Theta$. We use the Differentiable Interpolation-based Renderer (DIB-R) [24] to approximate functions $\mathcal{R}_F$ and $\mathcal{R}_S$ in a differentiable manner. The camera is assumed to be static. Moreover, we define a mesh transformation function $\mathcal{M}(\Theta, \mathbf{r}, \mathbf{t})$ that rotates mesh vertices around their center of mass by a quaternion-encoded 3D rotation $\mathbf{r} \in \mathbb{R}^4$ and translates them by a 3D vector $\mathbf{t} \in \mathbb{R}^3$. Then, the generalized rendering-based image formation model $\hat{I}(\Theta, \mathbf{r}, \Delta\mathbf{r}, \mathbf{t}, \Delta\mathbf{t}; B)$ is defined by

$$\hat{I}(\cdot) = \int_0^1 \mathcal{R}_F\big(\mathcal{M}(\Theta, \mathbf{r}+\tau\cdot\Delta\mathbf{r}, \mathbf{t}+\tau\cdot\Delta\mathbf{t})\big)\mathrm{d}\tau + \Big(1 - \int_0^1 \mathcal{R}_S\big(\mathcal{M}(\Theta, \mathbf{r}+\tau\cdot\Delta\mathbf{r}, \mathbf{t}+\tau\cdot\Delta\mathbf{t})\big)\mathrm{d}\tau\Big)\cdot B \quad (4)$$

where time step $\tau = 0$ corresponds to the initial object pose at the beginning of the camera exposure time, and $\tau = 1$ corresponds to the final pose at the end. This image formation model generalizes all previous formation models for FMOs (1), (2), (3). In goes beyond by explicitly modeling the object's 3D shape and 3D motion. Thus, the parameters that describe the object and its motion are the motion representation $\{\mathbf{r}, \Delta\mathbf{r}, \mathbf{t}, \Delta\mathbf{t}\}$ and $\Theta$, which contains the mesh vertices offsets and the texture map.

## 3.1 Loss terms

**Image formation loss.** The main component of our method is the image formation loss that measures the input image reconstruction according to (4). It is the difference between the observed input image and its reconstruction by the rendering-based formation model:

$$\mathcal{L}_I(\Theta, \mathbf{r}, \Delta\mathbf{r}, \mathbf{t}, \Delta\mathbf{t}; I, B) = \|I - \hat{I}(\Theta, \mathbf{r}, \Delta\mathbf{r}, \mathbf{t}, \Delta\mathbf{t}; B)\|_1 \ . \quad (5)$$

We noticed that (5) is directly related to the quality of reconstruction and deblurring as measured by the evaluation metrics PSNR and SSIM on the FMO deblurring benchmark [11] (see supplementary). However, directly minimizing the loss (5) is challenging since it is under-constrained and with many local minima, as observed experimentally. Therefore, additional priors and regularizers are vital.

**Silhouette consistency loss.** Another important prior is that the rendered silhouettes should stay close to sub-frame masks $M_\tau$ estimated by DeFMO [11]. This guides the 3D shape estimation

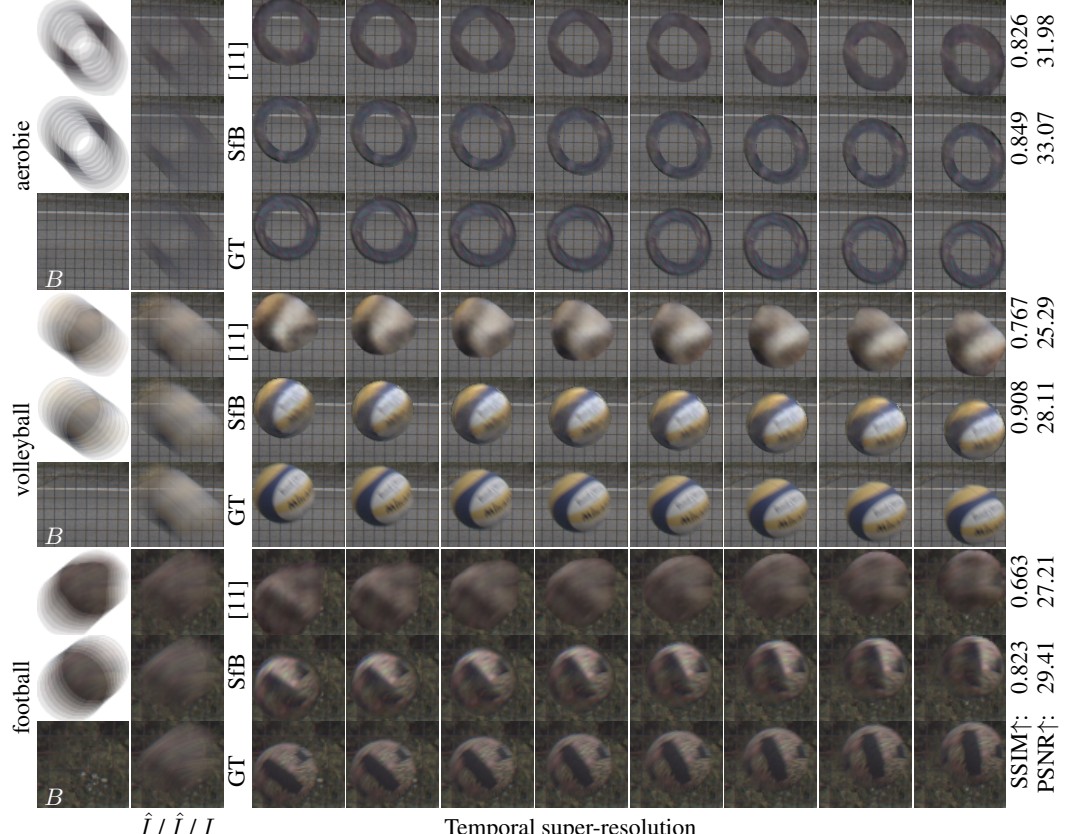

Figure 3: **Results on the TbD-3D [23] dataset.** We compare the proposed Shape-from-Blur (SfB) method with the previous state-of-the-art DeFMO [11] and also show the ground truth from a high-speed camera (GT). The actual input image $I$ (GT row) is almost indistinguishable from the image $\hat{I}$ ([11] and SfB row) rendered as a mixture of the background $B$ and the reconstructed object appearance, temporally averaged over the image exposure time (shown above the background).

with the approximate object location in the image, which helps especially in the early steps of the optimization process. The silhouette consistency is defined as

$$\mathcal{L}_S(\Theta, \mathbf{r}, \Delta\mathbf{r}, \mathbf{t}, \Delta\mathbf{t}) = 1 - \int_0^1 \text{IoU}\Big(M_\tau, \mathcal{R}_S\big(\mathcal{M}(\Theta, \mathbf{r} + \tau \cdot \Delta\mathbf{r}, \mathbf{t} + \tau \cdot \Delta\mathbf{t})\big)\Big)\, \mathrm{d}\tau \ , \quad (6)$$

where the Intersection over Union (IoU) is computed between the mesh silhouettes estimated by our method and the input DeFMO masks.

**Laplacian shape regularization loss.** Due to the 3D-to-2D projection, many meshes generate the same input image, especially in the mesh areas not visible in the image ('back sides' of the object). Thus, in order to favor smooth meshes, we add a commonly used [24, 26, 25] Laplacian shape regularization $\mathcal{L}_L(\Theta)$.

**Texture smoothness loss.** We favor smooth piecewise-constant textures by adding a total variation loss $\mathcal{L}_T(\Theta)$ on the estimated texture map, which is a sparse gradient prior [39], computed as the mean absolute difference between neighboring pixels in both image directions. Thanks to this term, the texture in unobserved mesh parts is a smooth transition between the edges of the observed parts. Without $\mathcal{L}_T(\Theta)$, our method produces random noise pixels in the unobserved texture parts, which degrade the quality of novel view synthesis.

**Joint loss.** The joint loss is a weighed sum of the image formation loss (5), the silhouette consistency (6), Laplacian shape regularization, and texture smoothness:

$$\mathcal{L} = \mathcal{L}_I(\Theta, \mathbf{r}, \Delta\mathbf{r}, \mathbf{t}, \Delta\mathbf{t}; I, B) + \mathcal{L}_S(\Theta, \mathbf{r}, \Delta\mathbf{r}, \mathbf{t}, \Delta\mathbf{t}) + \mathcal{L}_T(\Theta) + \lambda_L \cdot \mathcal{L}_L(\Theta) \ . \quad (7)$$

In Sec. 3.2, we infer the parameters $\{\Theta, \mathbf{r}, \Delta\mathbf{r}, \mathbf{t}, \Delta\mathbf{t}\}$ describing the object texture, shape, and motion by test-time optimization on the input image $I$ and background $B$.

**Technical details.** The weight of the Laplacian shape regularization is set to $\lambda_L = 1000$. There are no weights for other terms since the default unit weight produced good results. We use the Kaolin [40] implementation of DIB-R [24], and set the smoothness term for soft mask rasterization to $7000^{-1}$. We use a median of the previous five frames in a video as an approximation of the background.

## 3.2 Loss optimization

We optimize the joint loss (7) using ADAM [41] with the fixed learning rate 0.1 for 500 iterations in the PyTorch [42] framework. This is made possible by using a differentiable renderer, which enables to compute gradients of the rendered image w.r.t. the object parameters. Optimization takes about 60 seconds on a single 8 GB Nvidia GTX 1080 GPU (for 500 iterations of a single prototype mesh with 1212 vertices, 2420 faces, $100 \times 100$ texture map, 8 time steps to discretize the intergral, and input image of $240 \times 320$ pixels).

**Prototypes.** We provide the method with a set of three prototypes. The best prototype is automatically selected based on the lowest value of the image formation loss (5) after optimization starting from each prototype in turn. The first prototype is a sphere with 1212 vertices, 2420 faces, and simple spherical projection texture mapping. The second prototype is a bigger sphere with 1538 vertices, 3072 faces, and with more sophisticated Voronoi texture mapping [43]. The third prototype is a torus with a similar number of vertices and faces and the same Voronoi mapping. We used two spheres with different number of vertices to include more diversity in the complexity of the genus 0 shape. More vertices allow modeling more local shape details, e.g. concavities.

**Normalization and initialization.** The mesh vertices are normalized to zero center of mass and unit variance, maintaining the normalization throughout the whole optimization process. Normalization to unit variance is needed to keep the mesh in the canonical space. Without such normalization, it would be ambiguous whether the mesh vertices should be moved or the initial pose $\{\mathbf{r}, \mathbf{t}\}$ changed. Moreover, this normalization makes it easier to keep the object projection within the image bounds.

Texture values are normalized by Sigmoid activation. Translation vectors $\mathbf{t}$ and $\mathbf{t} + \Delta\mathbf{t}$ are estimated as a hyperbolic tangent (tanh) activation with normalization such that the 2D object projection is always visible in the image, i.e. $(-1, -1)$ is the bottom left corner of the image, $(1, 1)$ is the top right corner, a $z$ coordinate of $-1$ indicates an object covering the whole field-of-view (FOV), and $z = 1$ indicates covering $5\%$ of FOV. Rotations $\mathbf{r}, \Delta\mathbf{r}$ are represented by quaternions, and $\Delta\mathbf{r}$ is capped to represent at most 120 degrees total rotation. The mesh is initialized with a uniform grey texture, zero vertex offsets, zero rotation, zero translation, and placed in the center of the image. Experimentally, initializing the texture by projecting the blurry input image directly onto the mesh led to inferior performance. The reason might be that blurry object appearance is a local optimum for the texture, making it more difficult for the optimization process to converge to the sharp object appearance.

In sum, the proposed method estimates the object's textured mesh $\Theta$ and its motion $\{\mathbf{r}, \Delta\mathbf{r}, \mathbf{t}, \Delta\mathbf{t}\}$ by test-time optimization. The input image is explained by the object parameters via minimizing the joint loss (7) that consists of the image formation loss (5) and suitable regularizers.

## 4 Evaluation

**Datasets.** We evaluate our method on four datasets. Three real-world datasets from the FMO deblurring benchmark [11]: **(1)** TbD [18], with 12 sequences (471 frames total); **(2)** TbD-3D [23], with 10 sequences (516 frames); **(3)** Falling Objects [22], with 6 sequences (94 frames); **(4)** and a synthetic dataset from [11]. The easiest dataset is TbD [18] with uniformly colored, mostly spherical objects. TbD-3D [23] is more difficult and contains objects with complex texture and significant 3D rotation, but the objects are still mostly spherical. Falling Objects [22] is the most challenging dataset, with objects of complex 3D shape (e.g. key, pen, marker) and significant rotation. The ground truth for all three real-world datasets includes the high-speed camera footage at $8\times$ higher frame rate, as well as the sub-frame 2D object locations and 2D masks. We use the synthetic dataset to evaluate the 3D shape $\Theta$, 3D translation $\Delta\mathbf{t}$, and 3D rotation $\Delta\mathbf{r}$ estimated by our method, as there are no real-world datasets of blurred objects with 3D models annotations. The dataset is created by applying the rendering-based image formation model (4) to synthetically moving 3D ShapeNet [44]

Table 1: **Evaluation on FMO benchmark.** The best-performing method is highlighted. Generic deblurring methods [2, 3] do not estimate the object trajectory, thus their TIoU is not defined (N / A). The datasets are sorted from the most challenging one [22] to the easiest one [18].

| Method | Falling Objects [22] | | | TbD-3D Dataset [23] | | | TbD Dataset [18] | | | Time [s]↓ |
| | TIoU↑ | PSNR↑ | SSIM↑ | TIoU↑ | PSNR↑ | SSIM↑ | TIoU↑ | PSNR↑ | SSIM↑ | |
|---|---|---|---|---|---|---|---|---|---|---|
| Jin et al. [3] | N / A | 23.54 | 0.575 | N / A | 24.52 | 0.590 | N / A | 24.90 | 0.530 | 0.5 |
| DeblurGAN [2] | N / A | 23.36 | 0.588 | N / A | 23.58 | 0.603 | N / A | 24.27 | 0.537 | 0.1 |
| TbD [18] | 0.539 | 20.53 | 0.591 | 0.598 | 18.84 | 0.504 | 0.542 | 23.22 | 0.605 | 100 |
| TbD-3D [23] | 0.539 | 23.42 | 0.671 | 0.598 | 23.13 | 0.651 | 0.542 | 25.21 | **0.674** | 1000 |
| DeFMO [11] | 0.684 | 26.83 | 0.753 | 0.879 | 26.23 | 0.699 | 0.550 | 25.57 | 0.602 | 0.05 |
| SfB (ours) | **0.701** | **27.18** | **0.760** | **0.921** | **26.54** | **0.722** | **0.610** | **25.66** | 0.659 | 1969 |

objects. The backgrounds are randomly sampled from sequences in the VOT dataset [45]. Rotation and translation are generated randomly, with an average translation of $3\times$ the object size and an average rotation of $100°$. This way, we generate 10 images, one for each of 10 different objects.

**Evaluation metrics.** We evaluate the deblurring accuracy by comparing the sub-frame object reconstruction to high-speed camera frames. We use Trajectory-IoU (TIoU) [18], which averages the standard IoU between the estimated silhouette and the ground truth mask along the ground truth trajectory. For evaluating sub-frame appearance, we generate a temporal super-resolution image sequence by applying the rendering-based image formation model (4) at a series of time sub-intervals (changing the integration bounds) using the object parameters $\{\Theta, \mathbf{r}, \Delta\mathbf{r}, \mathbf{t}, \Delta\mathbf{t}\}$ estimated by our method. We then compare this generated sequence to the ground truth high-speed frames by the Peak Signal-to-Noise Ratio (PSNR) and Structural Similarity (SSIM) metrics.

Next, we evaluate the error in estimating the 3D translation $\Delta\mathbf{t}$ and 3D rotation $\Delta\mathbf{r}$ that the object undergoes during the camera exposure time. The translation error is measured as $\epsilon_{\Delta\mathbf{t}} = \|\Delta\mathbf{t} - \Delta\mathbf{t}^{\text{GT}}\|_2$, and the rotation error $\epsilon_{\Delta\mathbf{r}}$ is the angle between $\Delta\mathbf{r}$ and $\Delta\mathbf{r}^{\text{GT}}$ (GT denotes ground truth values). Note that the canonical mesh representation of ShapeNet objects is different from the one estimated by our method, preventing us from evaluating the estimate of the initial 6-DoF pose $\{\mathbf{r}, \mathbf{t}\}$.

To measure the 3D shape reconstruction accuracy, we align the estimated $\Theta$ and ground truth $\Theta^{\text{GT}}$ meshes and compute the average bidirectional distance between them as $\epsilon_\Theta = {}^1\!/_2(\text{D}(\Theta, \Theta^{\text{GT}}) + \text{D}(\Theta^{\text{GT}}, \Theta))$, where $\text{D}(\Theta_1, \Theta_2)$ is the average distance of each vertex of $\Theta_1$ to the closest face of $\Theta_2$. The meshes are rescaled to tightly fit a unit cube, thus a unit of $\epsilon_\Theta$ and $\epsilon_{\mathbf{t}}$ is one object size (diameter).

### 4.1 Comparison to previous methods on TbD, TbD-3D, and Falling Objects datasets

We compare to two generic deblurring methods, DeblurGAN-v2 [2] and Jin et al. [3]. We also compare to methods specialized for FMO deblurring, which are the TbD method [18], TbD-3D method [23], and the state-of-the-art DeFMO [11].

Our SfB method outperforms the above methods on all metrics on the two most challenging datasets Falling Objects and TbD-3D, setting the new state-of-the-art (Table 1). On the easier TbD dataset, SfB achieves the highest TIoU and PSNR, but marginally lower SSIM than TbD-3D, which is specifically designed for uniformly colored spherical objects. We show qualitative results in Fig. 3 and Fig. 4.

### 4.2 Ablation study on Falling Objects and synthetic datasets

The **first** block in the ablation study (Table 2) shows that omitting the silhouette consistency loss $\mathcal{L}_S$ (6) degrades performance by a wide margin, highlighting its importance. We also experiment with an alternative form of the silhouette consistency loss, which uses simple background subtraction instead of the DeFMO masks within (6). More precisely, we replace $M_\tau$ with the difference $D$ between the input image and the background, thresholded at 0.1 (details in supplementary). Using $\mathcal{L}_D$ is better than completely omitting any silhouette consistency, but performs worse than $\mathcal{L}_S$, since DeFMO localizes the object with the sub-frame precision, thereby providing a strong signal to our method. We also tried another loss $\mathcal{L}_F$ that favors the textures rendered by $\mathcal{R}_F$ to be similar (in terms of L2 loss) to the sub-frame appearances $F_\tau$ estimated by DeFMO. As shown in Table 2, this loss

Table 2: **Ablation study**. We evaluate variants of the silhouette consistency loss (6), the influence of the number of prototypes, of using the Laplacian and texture smoothness terms, and of the number of time steps used for discretizing integrals in (4). The best setting uses all proposed loss terms, all prototypes, and $5 \times 8$ discretization steps, where 8 is the number of ground truth high-speed frames.

| | Version | Falling Objects dataset [22] | | | | Synthetic 3D dataset [11] | | | Time [s]↓ |
|---|---|---|---|---|---|---|---|---|---|
| | | $\mathcal{L}_I \downarrow$ | TIoU↑ | PSNR↑ | SSIM↑ | $\epsilon_{\Delta \mathbf{t}} \downarrow$ | $\epsilon_{\Delta \mathbf{r}} \downarrow$ | $\epsilon_{\Theta} \downarrow$ | |
| Silhouettes | SfB (no $\mathcal{L}_S$) | 0.0371 | 0.341 | 17.94 | 0.594 | 2.372 | 36.0° | 0.039 | 58 |
| | SfB ($\mathcal{L}_D$) | 0.0264 | 0.591 | 20.71 | 0.680 | 1.608 | 23.9° | 0.030 | 58 |
| | SfB ($\mathcal{L}_S$ and $\mathcal{L}_F$) | 0.0302 | 0.674 | 25.64 | 0.734 | 0.553 | 20.2° | 0.024 | 63 |
| | SfB ($\mathcal{L}_S$) | **0.0234** | **0.693** | **26.30** | **0.745** | **0.305** | **17.1°** | **0.023** | 63 |
| Terms | SfB (no $\mathcal{L}_L$) | 0.0269 | 0.676 | 25.84 | 0.735 | 0.462 | **14.2°** | **0.023** | 61 |
| | SfB (no $\mathcal{L}_T$) | 0.0276 | 0.690 | 24.98 | **0.751** | **0.286** | 17.5° | **0.023** | 62 |
| | SfB (all terms) | **0.0234** | **0.693** | **26.30** | 0.745 | 0.305 | 17.1° | **0.023** | 63 |
| Prototypes | SfB (torus) | 0.0309 | 0.677 | 25.59 | 0.736 | 1.741 | 21.3° | 0.032 | 63 |
| | SfB (sphere, no rot.) | 0.0298 | 0.691 | 25.84 | 0.737 | 0.835 | 20.4° | 0.024 | 61 |
| | SfB (sphere) | 0.0234 | 0.693 | 26.30 | 0.745 | 0.305 | 17.1° | 0.023 | 63 |
| | SfB (2 prototypes) | 0.0220 | 0.695 | 26.43 | 0.744 | 0.304 | 13.2° | 0.023 | 124 |
| | SfB (3 prototypes) | 0.0219 | **0.701** | 26.57 | **0.748** | **0.194** | **11.1°** | 0.023 | 241 |
| | SfB (+ 4 distractors) | **0.0215** | 0.696 | **26.65** | **0.748** | 0.250 | 15.7° | **0.022** | 421 |
| Time steps | SfB (1 × 8 steps) | 0.0234 | 0.693 | 26.30 | **0.745** | 0.305 | 17.1° | 0.023 | 63 |
| | SfB (3 × 8 steps) | 0.0212 | 0.690 | 26.36 | 0.738 | 0.320 | 19.5° | **0.021** | 158 |
| | SfB (5 × 8 steps) | **0.0209** | **0.698** | **26.39** | 0.736 | **0.293** | **17.0°** | 0.023 | 253 |
| | SfB (10 × 8 steps) | 0.0210 | **0.698** | 26.25 | 0.734 | 0.329 | 18.1° | 0.022 | 502 |
| | SfB (best of all) | **0.0191** | **0.701** | **27.18** | **0.760** | **0.194** | **11.0°** | **0.024** | 1969 |

degrades the performance since DeFMO appearances are still blurry, noisy, and the set of $F_\tau$ does not correspond to a consistent 3D object (Fig. 3).

The **second** block of Table 2 shows the improvements of the Laplacian shape regularization and texture smoothness losses on the Falling Objects dataset. However, since the shapes and textures in the synthetic dataset are significantly more complex (e.g. guitar, jar, skateboard, sofa, bottle, pillow), these losses increase the 3D errors by a small margin.

The **third** block demonstrates the benefits of using more prototypes. In order to test the robustness of the automatic prototype selection, we also include additional distractor prototypes (teapot, teddy, bunny, cow) that are not similar to other objects in the datasets. Adding them changes the performance only marginally since they are rarely selected, demonstrating robustness. Besides, the first two rows of this block also show that ignoring the rotation leads to a drop in performance.

The **last** block evaluates step count used for discretizing the integral in the rendering-based image formation model (4). The performance increases with the number of steps, until it saturates at around $5\times$ more steps than the number of ground truth high-speed frames available for each image (i.e. 8).

According to the ablation study on the Falling Objects dataset (Table 2), the best performing version of SfB is using all loss terms (including DeFMO masks in $\mathcal{L}_S$), 7 prototypes (including 4 distractors), and $5\times$ more integration steps than the desired temporal super-resolution output. This version takes 30 minutes optimization time due to the high number of prototypes and time steps, which improve performance only marginally. While we can run our method in a configuration taking only a few minutes, optimization time can still be seen as a limitation. We believe that faster runtime could be achieved by avoiding unnecessary computations and iterations, and leave this as future work.

### 4.3 Applications to super-resolution and novel view synthesis

The main application of SfB is temporal super-resolution. For videos of motion-blurred objects, SfB can generate a video of arbitrary higher frame rates by correctly modeling the object's 3D shape and motion (Fig. 3). The modeling of textured 3D shape allows to generate novel views of the fast moving object (Fig. 6). SfB correctly recovers the 3D shape and deblurred texture of various moving objects captured by a mobile device (Fig. 4, boots, tape roll) or found on YouTube (Fig. 4, tennis ball).

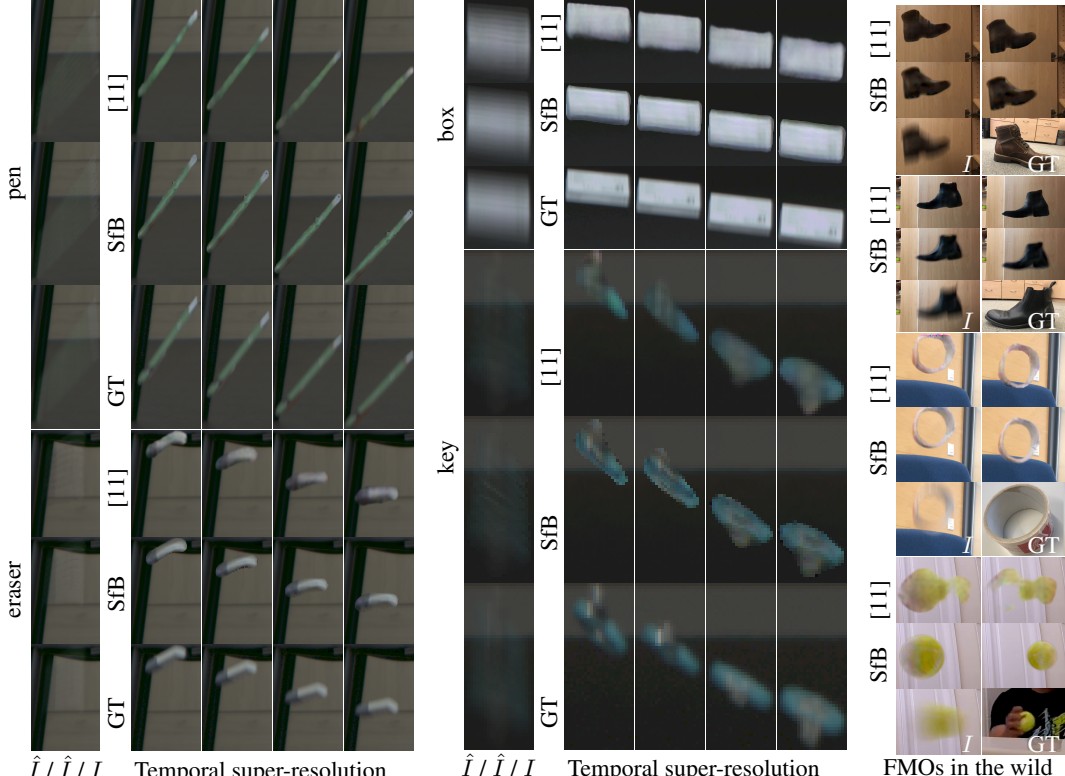

Figure 4: **Qualitative results for temporal super-resolution. Left and middle:** Falling Objects dataset [22]. **Right:** FMOs in the wild [11], featuring thrown objects captured by a mobile device camera or objects found on YouTube.

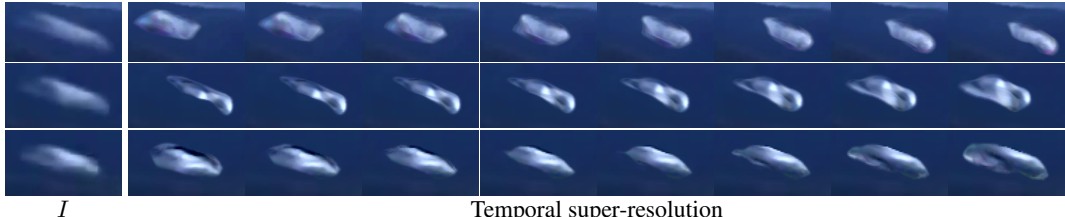

Figure 5: **Influence of image noise and compression artifacts.** Input images of a blurred flying bird are sorted from the least noisy (top) to the most noisy (bottom). Since compression and image noise are not modelled, they slightly deteriorate the reconstruction quality.

3D object rotation estimation has applications in sports analysis, e.g. in professional tennis games. For known intrinsic camera parameters, velocity and rotation can be estimated in real-world units.

## 4.4 Limitations

Since we assume a static camera, the method is not designed to deal with blur induced by a moving or shaking camera. Experimentally, we observed that the method is robust to small amounts of moving camera blur. Among other limitations are long optimization time, rigid body assumptions, and inconsistent estimation of unobserved object parts.

The proposed image formation model does not model any type of image noise, e.g. additive image noise, JPEG artifacts, compression, and others. In general, these factors can deteriorate the performance of the method (Fig. 5). However, the minimization of the reprojection error suppresses noise, and even the basic image formation model without the image noise still leads to satisfactory results in many real-world scenarios, as shown in the experimental section.

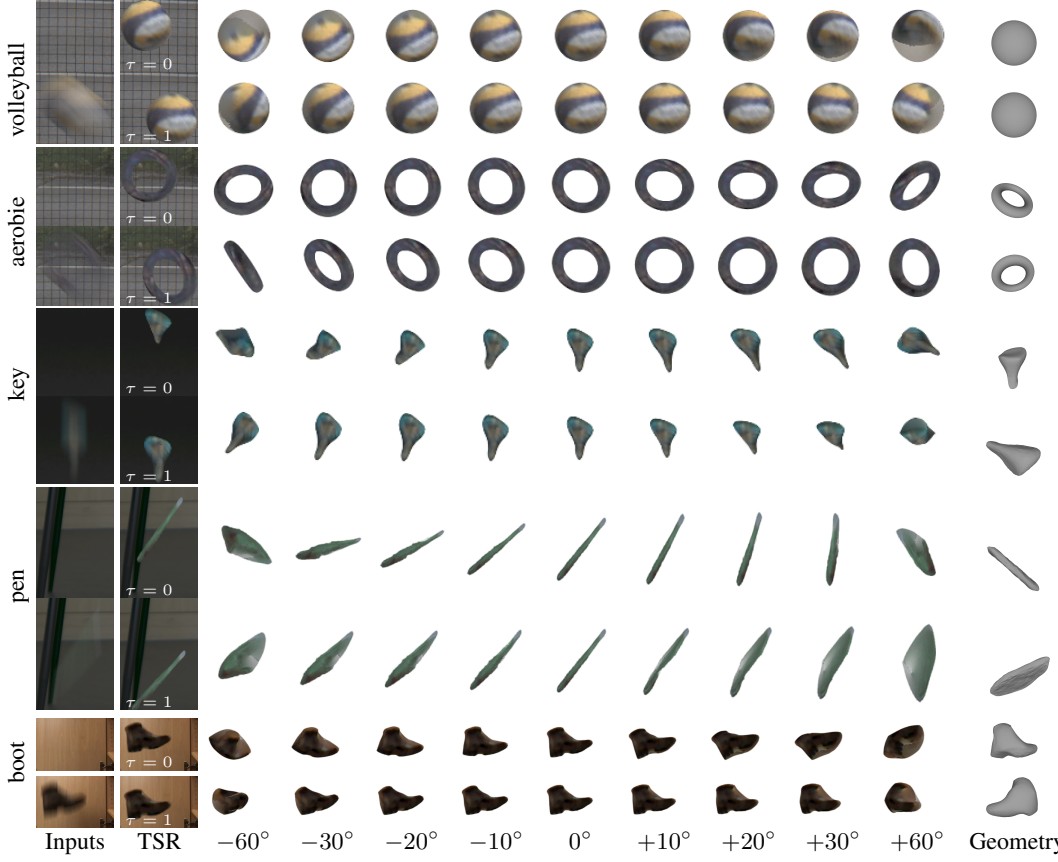

Figure 6: **Novel view synthesis.** From a single blurry image (1st column), our method generates novel views of the objects in the range $-60°$ to $+60°$ around the input view. We also show temporal super-resolution (TSR) at $\tau \in \{0, 1\}$ and the reconstructed 3D shape without texture (Geometry).

## 5    Conclusions and future work

We introduced and tackled a novel task to estimate the textured 3D shape and motion of fast moving objects given a single image with a known background. The proposed method is the first solution to this problem, and also sets a new state of the art on 2D fast moving object deblurring. Since the shape estimator deforms the (often spherical) prototype shape, the shape often remains unchanged along unobserved directions (Fig. 6, pen). To address some of the limitations, future work includes adding shape priors, such as symmetry or data-driven priors, and more powerful shape representations for expressing arbitrary shape topologies. The differentiable SfB loss can be incorporated into deep learning pipelines to benefit from large training datasets. SfB can be generalized to optimize over multiple video frames, rather than just one. For instance, when optimizing the textured 3D shape for several consecutive frames, the object becomes visible from more viewpoints providing more constraints and fewer ambiguities in unobserved parts.

## Acknowledgments and Disclosure of Funding

This work was supported by a Google Focused Research Award, Innosuisse grant No. 34475.1 IP-ICT, and a research grant by the International Federation of Association Football (FIFA).

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
