# Shape from Blur: Recovering Textured 3D Shape and Motion of Fast Moving Objects
## – Supplementary Material –

**Denys Rozumnyi**
Department of Computer Science
ETH Zurich, Switzerland
denys.rozumnyi@inf.ethz.ch

**Martin R. Oswald**[1,2]
[2]University of Amsterdam
Netherlands
martin.oswald@inf.ethz.ch

**Vittorio Ferrari**
Google Research
Zurich, Switzerland
vittoferrari@google.com

**Marc Pollefeys**
[1]Department of Computer Science
ETH Zurich, Switzerland
marc.pollefeys@inf.ethz.ch

## A   Loss terms

This section provides additional details about the formulation of the loss terms.

**Texture smoothness loss.**   The texture smoothness loss $\mathcal{L}_T(\Theta)$ is implemented by total variation. If the number of pixels is $K$, and the derivative of the estimated texture map is $T_x$ in $x$ image direction and $T_y$ in $y$ image direction, then the loss is defined as

$$\mathcal{L}_T(\Theta) = \frac{1}{K} \sum_{p=1}^{N} |T_x(p)| + |T_y(p)| \ . \tag{1}$$

**Laplacian shape regularization loss.**   To define the Laplacian shape regularization loss $\mathcal{L}_L(\Theta)$, we follow the notation proposed in the Soft Rasterizer method [1]. We assume that the mesh $\Theta$ consists of a set of vertices, where each vertex $v_i$ is a 3-dimensional vector. Then, the function $N(v_i)$ represents the neighbors of vertex $v_i$: a set of all adjacent vertices as defined by the faces in $\Theta$. The loss is the sum of differences between each vertex and the center of mass of all its neighbors:

$$\mathcal{L}_L(\Theta) = \sum_i \left\| v_i - \frac{1}{N(v_i)} \sum_{j \in N(v_i)} v_j \right\|_2^2 \ . \tag{2}$$

**Intersection over Union.**   We clarify the definition of the Intersection over Union (IoU), which we use in the silhouette consistency loss in the main paper. The IoU between two masks (real-valued) or silhouettes (binary) is defined as

$$\text{IoU}(M_1, M_2) = \frac{\|M_1 \cdot M_2\|_1}{\|M_1 + M_2 - M_1 \cdot M_2\|_1} \ . \tag{3}$$

**Losses in the ablation study.**   In the ablation study, we experimented with an alternative form of the silhouette consistency loss, which is the difference image-based loss $\mathcal{L}_D$. It uses simple background subtraction instead of sub-frame DeFMO [2] masks. The difference image is binarized as $D = \|I - B\|_2 > 0.1$. The loss is implemented as the intersection over union (IoU) between $D$ and the estimated silhouettes, averaged over the exposure duration, and binarized:

$$\mathcal{L}_D(\Theta, \mathbf{r}, \Delta\mathbf{r}, \mathbf{t}, \Delta\mathbf{t}) = 1 - \text{IoU}\left( D, \int_0^1 \mathcal{R}_S\big(\mathcal{M}(\Theta, \mathbf{r} + \tau \cdot \Delta\mathbf{r}, \mathbf{t} + \tau \cdot \Delta\mathbf{t})\big) \, d\tau > 0 \right) \ . \tag{4}$$

35th Conference on Neural Information Processing Systems (NeurIPS 2021).

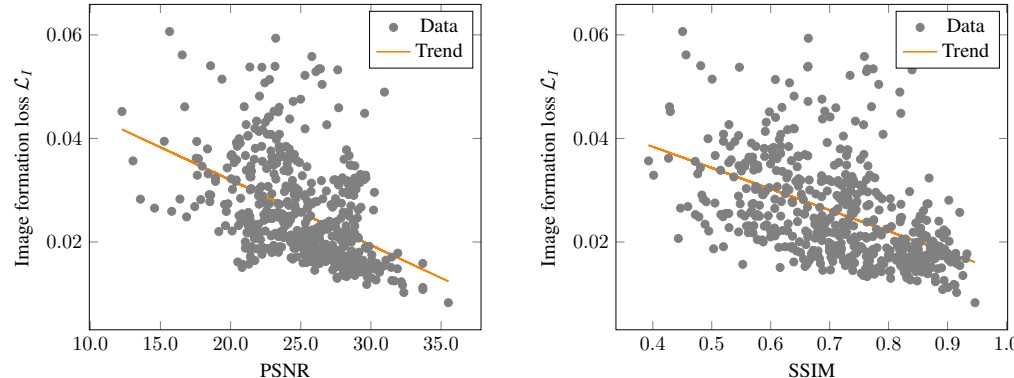

Figure 1: **Correlation between the image formation loss and evaluation metrics (PSNR/SSIM).** Linear regression of the measured data shows the trend: the lower is the image formation loss, the better are the evaluation metrics. Thus, it can be viewed as a surrogate for the evaluation metrics.

We also experimented with another loss $\mathcal{L}_F$, favoring the textures rendered by the proposed method to be similar to the sub-frame appearances estimated by DeFMO. The mathematical form is similar to the silhouette consistency loss, but instead of the masks, it directly compares the RGB values:

$$\mathcal{L}_F(\Theta, \mathbf{r}, \Delta\mathbf{r}, \mathbf{t}, \Delta\mathbf{t}) = \int_0^1 \|F_\tau - \mathcal{R}_F\big(\mathcal{M}(\Theta, \mathbf{r} + \tau \cdot \Delta\mathbf{r}, \mathbf{t} + \tau \cdot \Delta\mathbf{t})\big)\|_1 \, \mathrm{d}\tau \ . \tag{5}$$

## B  Correlation between image formation loss and evaluation metrics

As mentioned in the main paper, we noticed that the image formation loss is directly related to the quality of reconstruction and deblurring as measured by the official evaluation metrics PSNR and SSIM on the FMO deblurring benchmark [2]. Fig. 1 shows all measured data over all input image sequences in all three datasets from the FMO benchmark. The orange line demonstrates the observed linear trend.

## C  Additional qualitative results

We show additional results in Fig. 2 for the TbD-3D dataset [3] and in Fig. 3 for the Falling Objects dataset [4].

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

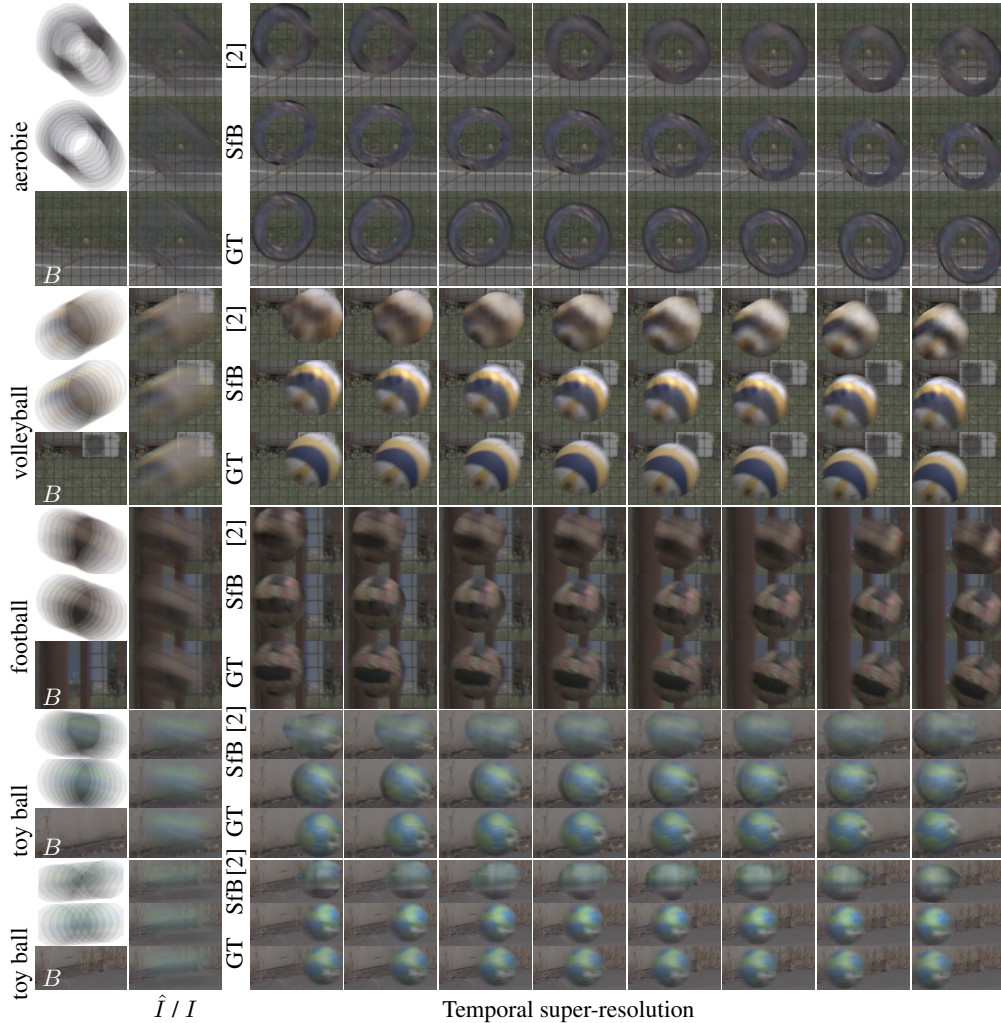

Figure 2: **Additional results on the TbD-3D [3] dataset.** We compare the proposed Shape-from-Blur (SfB) method with the previous state-of-the-art DeFMO [2], and also show the ground truth from a high-speed camera (GT). The actual input image $I$ is almost indistinguishable from the image $\hat{I}$ rendered by SfB as a mixture of the background $B$ and the reconstructed object appearance, temporally averaged over the image exposure time (shown above the background).

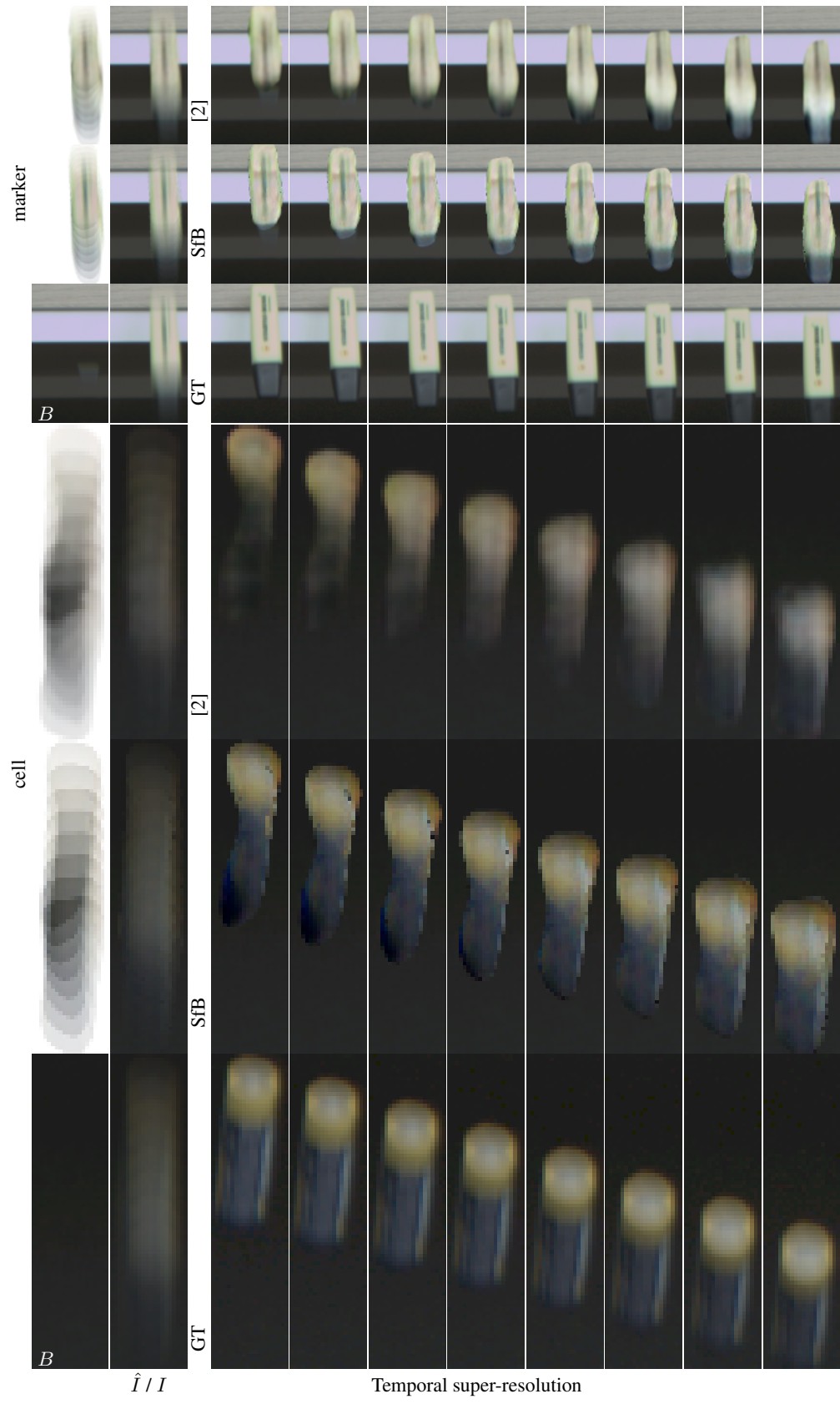

Figure 3: **Additional results on the Falling Objects [4] dataset.**