# OpenReview forum: "Shape from Blur: Recovering Textured 3D Shape and Motion of Fast Moving Objects"
_NeurIPS.cc/2021/Conference — NeurIPS 2021 Poster_

### Official Review · Reviewer_QNgV · 2021-07-16

**Rating:** 7
**Confidence:** 4

**Summary:**

This paper presents a method for 3D shape reconstruction of objects with large motion blurs in a video sequence. The authors propose to augment an off-the-shelf 2D shape-from-blur method, whose output is used to fit a 3D mesh shape as well as a linear 3D motion trajectory within a small timeframe. The motion blur is synthesized by averaging the shape & texture predictions within the timeframe for the image reconstruction loss, in addition to other regularization terms to keep the optimization well-behaved. Temporal super-resolution can also be achieved by synthesizing the image at a continuous timestamp. Experiments are benchmarked on 3 real-world datasets containing sequences of small objects and a synthetic dataset for benchmarking reconstruction accuracy.

**Limitations And Societal Impact:**

Adequately addressed

**Main Review:**

Strengths:
+ The paper is very nicely presented. The introduction and related work sections give a good overview of the importance of the problem of shape estimation from blurred images and clearly explain where this paper stands in the problem scope.
+ The paper proposes a simple and neat way of combining 2D shape-from-blur methods with differentiable mesh renderers to further predict the 3D shape structures. The problem formulation and method descriptions are also well-explained.
+ The 3D shape predictions are successful in most cases. The experiments show a clear advantage of how 3D prediction and modeling explicit 3D transformations help synthesize higher quality images for temporal super-resolution as well. Sufficient ablation studies are also provided for understanding the contribution from each component.

Weaknesses:
- I think it would be much more helpful if the method could be evaluated on datasets with larger objects and of higher resolutions. I'm not sure if this was intended to target low-resolutional images so that motion blurs are more significant, but the objects evaluated with seem to be of very low resolution, and it is difficult to visually compare the results (e.g. the temporal super-resolution results in Fig. 4 are not very informative, as everything is rather still blurry). I personally think only the volleyball, tape, and shoe examples are more discriminative.
- The explanation of prototypes is not very clear. Do the authors mean different shape templates with different genus numbers? It is not clear why spheres of different sizes (L171-176) are considered two different prototypes (shape templates), as I would expect the silhouette loss to find the right scale automatically. Also, it is not clear what "distractor prototypes" are (L247). My understanding is that a torus and a teapot would both be genus-1 and the rest genus-0.
- As all the objects considered in the paper are either genus-0 or 1, it would be helpful to discuss how the proposed framework (including DeFMO) could be extended to arbitrary shape topologies.
- As I understand, DeFMO also predicts the image appearance in addition to the silhouettes. Why isn't the predicted appearance utilized instead of the proposed mesh texture prediction? I would think it should make the optimization problem much more constrained as well.

Also, as the 3D transformations considered are linear with time (within the small timeframe), I think it would be helpful if the authors could provide some thoughts on whether such 3D motion blur could be more analytically synthesized without having to average multiple rendered samples across time, so as to more faithfully approximate the actual integral.

Other minor problems:
- The paragraphs in L242-245 and L246-249 refers to the wrong blocks and should be swapped.
- What does $\hat{I}/I$ in Fig. 3, 4 stand for? It's not clear which images correspond to $\hat{I}$ and $I$ respectively.
- The claim in L103 of estimating 3D shape from a single image sounds invalid and contradicts with L106 (background is also assumed given).

**Time Spent Reviewing:**

6hr

---

> ### Author Response · Authors · 2021-08-10
> **Response to Reviewer QNgV**
>
> Thank you for your positive review and constructive comments, which we address below.
> * **Dataset size and resolution.** We agree that richer and higher quality datasets are in need in this still-emerging field of FMOs. All available datasets with FMOs have already been used in the presented paper. Please also note that the actual image size is around 960 by 400 pixels in the FMO benchmark datasets, just the objects are typically depicted smaller. In fact, they have to be small enough to fulfill the FMO definition, which states that the object travels a distance larger than its size in one image frame.
> * **Explanation of prototypes.** Most objects in the FMO benchmark datasets have spherical topology (genus-0). Only the aerobie and the key have genus-1. Therefore, we used two spheres with 1212 vertices and 1538 vertices respectively to include more diversity in the complexity of the genus-0 shape. This is not about the right scale, but rather about modeling more local shape details (like concavities) with more vertices, e.g. a shoe or a pen. Spherical topology (genus-0) can also be expressed with only 4 vertices (tetrahedron), but such a prototype will be an extremely poor approximation for modeling any 3D object. **Distractor prototypes** were used to test how the method performs when the prototype mesh is far from the ground truth. In most cases, such distractor prototypes will either converge to the same correct mesh or are just not chosen based on the image formation loss. But we agree that they mostly have the same genus number. We will add more explanations and intuitions on the choice of prototypes in the camera-ready. Future work might look into more general solutions for selecting the right topology and shape complexity automatically.
> * **Arbitrary shape topologies.** Improved shape representation is probably one of the most interesting extensions of this work. One possible extension is to learn a low-space latent encoding via Variational Autoencoders (VAE) of the space of prototypes with different topologies. A more modern approach might be applying NeRF-like representation and rendering.
> * **DeFMO appearance.** We point out that we have in fact tried to utilize the DeFMO RGB appearances in the optimization. As stated in the ablation study in Table 2, SfB ($\mathcal{L}_S$ and $\mathcal{L}_F$, defined in the supplementary material L21), this leads to a drop in performance. Actually, this is expected “since DeFMO appearances are still blurry, noisy, and the set of $F_\tau$ does not correspond to a consistent 3D object” (L238-L241). Fig. 3 demonstrates a clear example of why DeFMO appearances actually decrease the performance.
> * **Analytical computation of motion blur.** This is indeed a very exciting idea on how to avoid integral approximation, but we are unfortunately not aware of any way to analytically model the motion blur via differentiable mesh rendering. This is an interesting direction for future work.
>
> Thank you very much for spotting minor problems. We will swap the paragraphs in L242-245 and L246-249, rephrase the sentence in L103, and clarify the difference between the input image $I$ and the synthetically generated “image $\hat{I}$ rendered by SfB as a mixture of the background $B$ and the reconstructed object appearance, temporally averaged over the image exposure time” (Figure 3 caption). We will make it more clear which image corresponds to which variable.

---

### Official Review · Reviewer_qzSX · 2021-07-16

**Rating:** 6
**Confidence:** 4

**Summary:**

The submitted paper focuses on the shape, texture, and motion recovery of a single fast moving object from two input images from a static camera - a clean background image, and a blurred image due to fast object motions. The authors tackle the problem using differential rendering and iterative optimization at test time. Specifically, the template mesh ("prototype"), the texture map, the initial translation and rotation, as well as the constant motion speed are jointly optimized in order to minimize a combination of loss functions, with the image formation loss being the key contribution. The method is demonstrated on both synthetic dataset and relatively simple real datasets.

**Limitations And Societal Impact:**

Yes

**Main Review:**

Originality

The proposed method is an application of differential rendering in the context of deblurring - several images are rendered separately and then integrated to form a blurred image, and then loss is back-propagated to rectify the shape, texture, and motion model. To the best of my knowledge, this proposed method differs from most of the recent SotA methods for deblurring, which focus on supervised learning and training on a large dataset.


Quality

The proposed method exceeds several prior work on the FMO benchmark. Based on the videos and images shown, the reconstructed mesh is reasonably close to the ground truth (considering that this is a heavily under-constrained problem, perfect reconstruction isn't possible). However, I suspect that the improvement in accuracy over prior methods is largely attributed to the fact that the template mesh is quite close the the ground truth in shape. This is verified by the prototypes study in Table 2 on Synthetic 3D dataset - when the ground truth shapes become slightly more complicated, the performance gap is significant (0.305 vs. 0.194) when different number of prototypes is used. Additionally, it is unclear how import some of the optimized variables are (e.g., rotational speed, texture optimization) and how much they contribute to the final accuracy

I think a few ablation studies can greatly improve the quality of the paper.
1. an ablation study on different mesh topologies as prototypes (when only 1 prototype is used), ideally we see how the performance drops when the topology deviates from the ground truth (e.g., torus vs. sphere)
2. an ablation study on the contribution of rotational speed
3. an ablation study on the contribution of texture optimization, if the texture is initialized properly (e.g., by projecting the image pixels directly onto the mesh)
4. how does the method perform on "slow-motion" blur, especially compared to prior work?


Clarity

The paper is overall well written and easy to follow, except for a few things:
1. Why is normalizing to unit variance needed? This is in the "Normalization and initialization" paragraph under Section 3.2
2. How is the number of subframes and the time difference selected?


Significance

The proposed method is a proof-of-concept on how differential rendering and a 3D prior can help improve 2D deblurring.

**Time Spent Reviewing:**

3

---

> ### Author Response · Authors · 2021-08-10
> **Response to Reviewer qzSX**
>
> Thank you for your positive review and constructive comments. We thank you for providing new ideas for interesting ablation studies. We have performed some of them and will include them in the camera-ready version.
> 1. As shown in the table below, when optimizing with only the torus topology, the performance drops on the Falling Objects dataset, where most objects have spherical topology.
> 2. We believe that all optimized variables are important. The rotational speed can be omitted in cases when the object only translates without perceivable rotation, e.g. due to constant color appearance. But in general, modeling the rotational speed is beneficial. For instance, in Fig. 3, it would be impossible to reconstruct and deblur the objects (volleyball, football, aerobie) without modeling the rotational speed. Again, the performance drops when the rotational speed is not considered (table below).
> 3. Texture optimization is one of the key attributes for detecting rotation and deblurring. We believe that initializing the texture by projecting the image pixels directly onto the mesh is far from the desired solution since the input image pixels are a combination of blurry appearance and the background. Moreover, the exact object location is not known either and is part of the optimization. Experimentally, the table below shows that when the texture is initialized as the blurred appearance from the input, the performance drops compared to uniform initialization. The reason might be that blurry object appearance is a local optimum for the texture, and it is more difficult to converge to sharp object appearance.
> 4. Small-motion blur essentially steers our method towards the extreme case of single-sharp-image 3D reconstruction, which is not the focus of our work. However, it can be expected that the benefit of our model is smaller since as a consequence there are also smaller baseline changes to gain more information for 3D shape estimation. Also, unfortunately, the currently available FMO datasets do not include small motion changes.
>
> | SfB Version           | $\mathcal{L}_I$ ↓| TIoU ↑| PSNR ↑| SSIM ↑ |
> | -----------------------  | ---------- | -------- | -------- | ------- |
> | Sphere (Table 2)   | **0.0234** | **0.693** | **26.30** | **0.745** |
> | 1. Torus                 | 0.0309 | 0.677 | 25.59 | 0.736 |
> | 2. No rot.               | 0.0298 | 0.691 | 25.84 | 0.737 |
> | 3. Blurry tex. init.   | 0.0308 | 0.658 | 25.82 | 0.736 |
>
> In this table, we use only one prototype (sphere or torus), 8 time steps, and all loss terms. This table will be incorporated in Table 2, together with an additional experiment on the synthetic dataset.
>
> **Clarity**
> 1. Normalization to unit variance is needed to keep the mesh in the canonical space. Without such normalization, it will be ambiguous whether the mesh vertices should be moved or the initial pose ($r,t$) changed. Also, this normalization makes it easier to keep the object projection in the image domain.
> 2. The number of subframes is selected such that it is as small as possible for computational reasons, but also large enough to discretize the integral. In Table 2, we ablated 4 settings (8, 24, 40, 80 time steps / number of subframes). As expected, the larger the number of time steps, the better are the scores (L251), but the runtime becomes slower. For most scenarios, 8 time steps (number of subframes) provide a good approximation.

---

> > ### Comment · Reviewer_qzSX · 2021-08-28
> > **Keeping original rating**
> >
> > I would like to thank the authors for their response, especially with the added experiments. It would be great to incorporate these clarifications into the updated version.
> >
> > I am keeping my original ratings because the method is fundamentally an example of test-time optimization using differential rendering, among many different variants of shape/texture representations in various applications. As is indicated by the new experimental results, the performance is highly dependent on using the correct shape prior, so it would be challenging to apply similar technique on more complex objects.

---

### Official Review · Reviewer_52ha · 2021-07-16

**Rating:** 9
**Confidence:** 5

**Summary:**

This paper presents SfB, a novel method of handling a single RGB image of a significantly blurred fast moving object, by recovering the object's shape, texture, and trajectory, what are optimized towards a joint loss of image consistencies, via a differentiable render. The method can be applied to a variant of tasks of de-blurring, shape reconstruction, super resolution and view synthesis. The paper reports state-of-the-art results on a variant of real-world and synthetic datasets, both quantitatively and qualitatively.

**Limitations And Societal Impact:**

1. Shape representation.
a) As the method relies on the limited set of fixed topology prototypes, it is hard to generalize to arbitrary or highly complicated shapes, as also stated in the paper. This is also a constraint from the used render, which requires such mesh format. Can the shape representation be improved to overcome this limitation(such as, can NeRF-like shape and render mechanism be used here?) Or can the prototype can reconstructed in the first place?
b) Rigid assumption. During the exposure time, the object is assumed to be of fixed shape, only changing its position by rigid transformation. It is not mentioned in the paper whether \Theta is allowed to change during the exposure time. Can this method be applied to non-rigid objects, such as clothes or moving parts? (For instance, can \Theta mesh representation dependent on time \t?)


**Main Review:**

While relying on some of the previous work's output, this paper presents a novel method that has not seen in related works. The formulation of this method is intuitive and clear, and the discussions are comprehensive.
The proposed method is robust against real-world inputs, and achieves state-of-the-arts results.
This paper is very well written, easy to follow, and contains all the details for re-implementing. The author also provided their code in the supplement materials.
Though the reviewer believes the most weakness of this paper comes from its shape representation(see Limitations), it does not void the core idea of retrieving 3D from blurred images, and further study is encouraged along the track of improving 3D reconstructions.
Overall, it is a strong paper that would be a good contribution to the community.


**Time Spent Reviewing:**

2

---

> ### Author Response · Authors · 2021-08-10
> **Response to Reviewer 52ha**
>
> Thank you for your positive review and constructive comments, which we address below.
> * **Shape representation.** We observed that the majority of FMOs have a simple shape. The use of a small set of shape prototypes fits our target well and drastically reduces the solution space. We believe this provides a good trade-off between method generality and efficiency of our method. An improved shape representation is definitely an exciting and promising extension of this work. Possible choices are NeRF-like representation or even just a low-space latent encoding via Variational Autoencoder (VAE) of the space of prototypes with different topologies.
> * **Rigid assumption.** In our current model, $\Theta$ is not allowed to change during the exposure time. This will be clarified in the camera-ready submission. We believe that modeling non-rigid time-varying shapes in the proposed single-blurry-image scenario is an extremely difficult and largely under-constrained problem. This is definitely an interesting direction for future work, especially for objects that bounce off the ground and deform at that moment or clothes and moving body parts.

---

### Official Review · Reviewer_VMLD · 2021-07-25

**Rating:** 6
**Confidence:** 3

**Summary:**

This paper introduces an approach for predicting the textured 3D shape and motion from blurred video frames. The approach is based on the differentiable rendering that can predict shapes and image formations that can be regarded as a combination of background and foreground scenes. The paper shows interesting results that can reasonably recover shapes, motion, and a deblurred texture map.

**Limitations And Societal Impact:**

Please refer to the questions in the main review.

**Main Review:**

The paper introduces an interesting approach to recover 3D shapes from challenging video frames. The approach is straightforward to understand and easy to follow. The overall pipeline that consists of mask prediction, object prediction, appearance prediction module seems to be adequately designed. The produced output is obviously superior to deblurring approaches as well as a concurrent approach called DeFMO [11].

Pros:

 + The paper proposed an approach that can jointly estimate the 3D shape, texture, and motion of an input object. The visual quality of the demonstrate results is quite impressive. It seems to be pretty hard to predict the shapes from the images, but the recovered scene is quite reasonable. It is also surprising that it is possible to recover shape and textures from such challenging input images. I am not fully following this area, so my judgment may not be accurate, but it is impressive.

 + The experiment is carefully designed to demonstrate the effectiveness of the proposed approach. The authors use the multiple evaluation metrics (L_I, TIoU, PSNR, SSIM) to evaluate the natural scenes and use translation or rotational error for the synthetic dataset. Given the constraint that cannot capture the ground truth of the shape (except for the synthetic case), the demonstration is reasonable. In addition, authors design self-baselines (such as L_d and L_f) to demonstrate the effectiveness of the proposed loss functions.

+ Authors provide the source code of the proposed approach. There was not enough time to go over the detailed validation of the submitted code as an emergency reviewer. However, such a submission could help to understand the proposed approach.

Cons:

 - It is not fully convincing the importance of the problem. In 26-29, “Our work could eventually make FMO tracking accessible to everyone using regular cameras rather than expensive high-speed cameras, or push the capabilities of current high-speed cameras to a new level…” I was expecting some challenging cases after reading the introduction, but the demonstrated image examples are pretty constrained. The camera stands still, and the single object is assumed to have constant rotational and translational velocity. Because of these, the demonstrated results look unrealistic to me. There are examples using the YouTube videos, but it is still not clear these are the natural scenes that “every people” (as stated in the introduction) may encounter. Rather than that, the YouTube examples seem to be carefully chosen by the authors. It is because the YouTube videos are pretty similar to the Falling Objects [21] and TbD-3D dataset [22], and TbD dataset [18]. The authors mention this issue as the limitation in Sec 5, but it does not entirely resolve the concern. How about just adding failure cases and include thorough discussions regarding them? This could bring a better understanding of the proposed approach.

- The proposed approach is a collection of well-known modules. The authors use DeFMO for the silhouette consistency prediction and differentiable interpolation-based render (DIB-R) [23] and Kaolin [39] for the image formation check. The approach combines these modules to solve the problem differently. During this procedure, there is no new module is proposed.

 - It is hard to understand the computational complexity of the proposed approach. Can authors compare the computation time with other approaches? Table 2 shows the time for the proposed approach with different configurations, but it does not tell how it is faster or slower than the baseline approaches. How about comparing the runtime with other approaches such as Jin et al. [3], DeblurGAN [2], TbD [18], TbD-3D [22], and DeFMO [11]?

- A sentence shown in lines 169-170 is quite misleading. It states, “input image of 240x320 pixels takes around *60* seconds…” This is a misleading statement since the best performing approach takes *1969* seconds, as shown in Table 2. The performance of the configuration that takes 30 mins is reported in Table 1, and the line 169-170 below Table 1 says it takes about a minute. This is how the paper can give the wrong message to readers. Instead, the configurations that take about 60 seconds are comparable to DeFMO [11], based on TIoU, PSNR, and SSIM as shown in Table 1 and Table 2. I suspect the differentiable rendering takes longer if the number of prototypes and the proposed loss terms are applied. Long optimization time could be the limitation of the proposed approach, and it should be properly mentioned in the paper.

- Regarding the three prototypes, it is not clear why the approach should have a small sphere and the bigger sphere. For readers, ‘big’ and ‘small’ is quite ambiguous since it does not provide any detailed information regarding the size of spheres. If the approach should have two kinds of spheres, why not using two different sizes of ‘torus’? This question and ambiguity make it hard to understand the intention of the prototypes.

- Is there any consideration on the additive image noise? The basic image formulation seems to ignore the effect of additive noise on the image. However, the target problem is to tackle challenging images and recover useful information, as highlighted in the introduction of the paper. Therefore, it is necessary to add a discussion of how the proposed approach could or could not handle challenging cases having noises. Such examples could be included in the limitation of the proposed approach (see my previous proposal regarding the limitations), and it could provide more information when the proposed approach works as expected or not.

Overall, the paper has several concerns, but this paper shows clear merits – the approach demonstrates the new pipeline can handle challenging blurred images of moving objects. I am leaning toward accepting for now. However, I am not an expert in this area, and I am not sure the NeurIPS venue would be the best fit for this problem. The confidence is not relatively high.

**Time Spent Reviewing:**

6 hrs

---

> ### Author Response · Authors · 2021-08-10
> **Response to Reviewer VMLD**
>
> Thank you for your positive review and constructive comments. We were initially also surprised by the quality of our results. These results confirm that proper physical modeling of the image formation under motion blur is a powerful concept. We address your comments below.
>
> * **Challenging data.** We show most results on data, for which the ground truth is available. Undoubtedly, there are many challenging in-the-wild scenarios that are not covered by such datasets since capturing ground truth is not an easy task. Therefore, we also showed results on YouTube videos and hand-held device footage (e.g. Fig. 4), which are already quite challenging. For the camera-ready, we will include additional results on some more challenging data and include thorough discussions about the cases when the proposed method fails, such as severe camera motion.
> * **Runtime comparison.** We report runtimes for other methods on the FMO benchmark datasets in the table below. Depending on the desired output accuracy, our runtimes are in the range of the TbD methods. This table will be included in the camera-ready.
>
>
> | Jin et al. [3] | DeblurGAN [2] | TbD [18] | TbD-3D [22] | DeFMO [11] | SfB (best) | SfB (fast) |
> | --------------- | -------------------- | ----------- | ---------------- | ----------------- | ------------- | ------------- |
> | 0.5 s            |   0.1 s              |  100 s     | 1000 s          | 0.05 s          |  1969 s      | 63 s         |
>
>
> * **Optimization time.** The sentence on L169-170 will be clarified to highlight that optimization time is around 60 s for a single prototype mesh and 8 time steps. The difference in performance between a configuration that takes several minutes and 30 minutes is marginal. The 30-minute version was used for ablating the influence of distractor prototypes (3+4) and many time steps (80), which are unnecessary and change the performance only marginally. Also, we believe that for faster runtime, the optimization can be further improved to avoid unnecessary computations and iterations. But we agree that currently, long optimization time can be a limitation of the proposed method. We will include this discussion in the camera-ready.
> * **Prototypes.** All three prototypes are defined in the paper (L174-L175) with the number of vertices, faces, and texture mapping. Additionally, the prototypes are submitted in the supplementary material (sfb_demo_code/prototypes/*.obj). We agree that the choice of prototypes is not obvious. We simply wanted to initialize the shape optimization with basic shapes that are close to the target object shapes. We will elaborate the explanation in the final version. Most objects in the FMO benchmark dataset (except for aerobie and key) have spherical topology (genus-0). Therefore, we used two spheres, “small sphere” with 1212 vertices (L173) and “big sphere” with 1538 vertices (L174), to include more diversity in the complexity of the shape. For example, for some objects such as a shoe or a pen, it is beneficial to have more vertices to model more local shape details like concavities. For future work, we look into more general solutions for selecting the right topology and shape complexity.
> * **Additive image noise.** Currently, we do not model any type of image noise. However, the minimization of the reprojection error suppresses noise, and even the basic image formation model without the image noise still leads to satisfactory results in many real-world scenarios, as shown in the experiments. The limitation section will be extended to discuss the challenging cases with a lot of image noise, e.g. additive image noise, JPEG artifacts, compression, and others. We agree that all of these factors can deteriorate the performance of the proposed method. Future work can include modeling different types of image noise for more robustness in challenging scenarios.

---

> > ### Comment · Reviewer_VMLD · 2021-08-30
> > **After reading author's feedback.**
> >
> > Thanks for the detailed feedback.
> >
> > I am generally satisfied with the feedback provided by the authors. Here are some thoughts.
> >
> > - It would be good to include some failure cases. In the current form, readers of this paper would not clearly catch when the approach successfully reconstructs the scene and which cases fail. As mentioned by the authors, I agree that severe camera motion would affect the result.
> > - The runtime and optimization time is somewhat long. This aspect should be clarified in the main paper, such as the discussion or conclusion section.
> > - Regarding the prototypes, please include the reasons why the three prototypes are designed. Including the discussion about the prototype selection would be interesting, as mentioned.
> > - I believe the images with noise and compression artifacts would be an excellent example to see the reconstruction ability of the proposed approach. It would be good to include such examples in the revision.
> >
> > In addition to that, could the approach handle such a case? For example, a ball is bouncing the ground (consider a tennis ball bouncing around the court's baseline). In that case, there would be an abrupt change of the motions, and the motion parameter prediction would be very challenging. I am curious since such cases would be an excellent example of the practical use of the proposed approach.
> >
> > I checked this point is already answered for 52ha reviewer, but such a case could be considered reconstructing a rigid object with abrupt motion change (although the ball should be deformed when bounced). Maybe authors may consider multi-stage optimization - first, the approach recovers shape and motion, and texture. In the next stage, some discrepancies due to motion change or shape deformation could be optimized further.
> >
> > I also checked other reviews. I generally agree with the points raised by other reviewers, and I think the authors properly answered the questions. It seems to be obvious the paper has the merit to be published to the community.
> >
> > Thanks,
> > Reviewer VMLD.

---

> > > ### Author Response · Authors · 2021-08-31
> > > **Additional feedback**
> > >
> > > Thanks for your suggestions for improving the paper quality. To summarize, we will include examples with the failure cases and on images with noise and compression artifacts. Discussions on the runtime and the selection of prototypes will also be added. From the points raised by other reviewers, we will incorporate the clarifications (as promised in individual responses) and include an ablation study suggested by Reviewer qzSX.
> > >
> > > In the presented constant speed formulation, the proposed approach cannot handle a case with an abrupt change of motion. However, if initially assuming a rigid body without deformation (as you suggested), it is a straightforward extension. The number of unknowns in the optimization will increase only by 3+4 (additional translational and rotational change), which is marginal compared to the number of unknowns in the mesh and texture representations.
> > >
> > > Thanks for the idea of multi-stage optimization. This is indeed a reasonable approach since the deformation usually happens only for a short period of time, and the shape and motion parameters could probably be correctly estimated even with an initial rigid body assumption. Then, an additional optimization for shape deformation and motion parameters near the bounce can be performed. We will look into it for future work.
> > >
> > > Thank you for your time, Authors

---

### Decision · Program_Chairs · 2021-09-27

**Decision:**

Accept (Poster)

**Comment:**

The main task of this work is fast motion deblurring and yet the paper claims to jointly estimate 3D shape and texture along with the motion of the object. The paper received positive reviews: 7, 9, 6, 6. The paper is written well with impressive results compared to previous works. However, the main limitations of the paper is the lack of novelty. As pointed out by a reviewer, the proposed approach is  an extension of the DeFMO paper. The authors use DeFMO for the silhouette consistency prediction and differentiable interpolation-based render (DIB-R) and Kaolin [39] for the image formation check. The approach combines these modules to solve the problem differently. While the DeFMO directly renders the appearance and silhouette directly from the given input image and estimated background, this work instead estimates a 3D shape from the given input image (and background) and then renders the appearance and silhouette.
It is also worthy to note that estimating 3D shape and differentiable rendering from an image have been previously studied [23, 24, 25, 26]. The fact that this work estimates it from a motion-blur image doesn't necessarily make it significantly different from previous works. Moreover, the 3D estimation from the motion-blur image is also constrained with specific types of prototypes which makes it less generalizable to all types of fast-moving objects.

The approach demonstrates a new pipeline for jointly estimating shape, texture, and motion with challenging blurred images of moving objects. I am not sure the NeurIPS venue would be the best fit for this problem. Based on the reviews and the author rebuttals, I lean toward to reject the paper.